# The Quenching of Long-Wavelength Fluorescence by the Closed Reaction Center in Photosystem I in *Thermostichus vulcanus* at 77 K

**DOI:** 10.3390/ijms252212430

**Published:** 2024-11-19

**Authors:** Parveen Akhtar, Ivo H. M. van Stokkum, Petar H. Lambrev

**Affiliations:** 1HUN-REN Biological Research Centre, Temesvári krt. 62, 6726 Szeged, Hungary; akhtar.parveen@brc.hu; 2Department of Physics and Astronomy and LaserLaB, Faculty of Science, Vrije Universiteit Amsterdam, De Boelelaan 1081, 1081 HV Amsterdam, The Netherlands; i.h.m.van.stokkum@vu.nl

**Keywords:** chlorophyll, cyanobacteria, energy transfer, light harvesting, *Thermosynechococcus vulcanus*, time-resolved fluorescence

## Abstract

Photosystem I in most organisms contains long-wavelength or “Red” chlorophylls (Chls) absorbing light beyond 700 nm. At cryogenic temperatures, the Red Chls become quasi-traps for excitations as uphill energy transfer is blocked. One pathway for de-excitation of the Red Chls is via transfer to the oxidized RC (P700^+^), which has broad absorption in the near-infrared region. This study investigates the excitation dynamics of Red Chls in Photosystem I from the cyanobacterium *Thermostichus vulcanus* at cryogenic temperatures (77 K) and examines the role of the oxidized RC in modulating their fluorescence kinetics. Using time-resolved fluorescence spectroscopy, the kinetics of Red Chls were recorded for samples with open (neutral P700) and closed (P700^+^) RCs. We found that emission lifetimes in the range of 710–720 nm remained unaffected by the RC state, while more red-shifted emissions (>730 nm) decayed significantly faster when the RC was closed. A kinetic model describing the quenching by the oxidized RC was constructed based on simultaneous fitting to the recorded fluorescence emission in Photosystem I with open and closed RCs. The analysis resolved multiple Red Chl forms and variable quenching efficiencies correlated with their spectral properties. Only the most red-shifted Chls, with emission beyond 730 nm, are efficiently quenched by P700^+^, with rate constants of up to 6 ns^−1^. The modeling results support the notion that structural and energetic disorder in Photosystem I can have a comparable or larger effect on the excitation dynamics than the geometric arrangement of Chls.

## 1. Introduction

Photosystem I (PSI) is one of the most efficient photosynthetic complexes in nature, converting solar energy into chemical energy with a quantum yield of near unity [1]. This remarkable efficiency is attributed to its highly organized structure and fast processes of excitation energy transfer (EET) and primary photochemistry. The PSI core complex is composed of more than 10 protein subunits and typically binds around 100 chlorophyll *a* (Chl *a*) molecules [2,3] in a fused antenna-reaction center (RC) pigment-protein complex. Electronic excitations of the Chls are rapidly transferred to the RC Chl *a* dimer, denoted as P_700_, which acts as a primary electron donor initiating the photoinduced electron-transport chain.

One unique aspect of the antenna complex of PSI in nearly all organisms is the presence of long-wavelength or “Red” Chls absorbing light at wavelengths longer than 700 nm, i.e., having excited-state energies below that of the RC [4]. While the presence of Red Chls extends the absorption of far-red light, the photochemical conversion of the long-wavelength radiation absorbed by the Red Chls is only possible through energetically uphill transfer to the RC [5,6]. This thermodynamic barrier slows down the overall EET and trapping, depending on the number of Red Chls present, their location, and absorption wavelengths [7,8]. The overall trapping time strongly correlates with the number and energy levels of the Red Chls—a higher number of red forms increases the probability that excitation will visit a red form. On the other hand, excitations reside for longer times in low-energy states. However, experiments by Russo et al. [9] have suggested that the location of Red Chls with respect to the RC may be less important than their energy.

After charge separation, PSI relies on mobile electron carriers, mainly plastocyanin, to re-reduce the oxidized primary donor P_700_^+^. Depending on the physiological state, photosystem II activity can be (transiently) diminished, leading to donor-side limitation of the PSI electron transport, wherein PSI with closed RCs containing P_700_^+^ are unable to perform photochemistry. Under such conditions, excitation energy can be localized preferentially on the Red Chls, which would increase the risk of photodamage if the excitation energy is not efficiently quenched. Fortunately, in PSI, P_700_^+^ is a potent excitation quencher [10] that is able to deactivate excited states in the antenna Chls on a similar time scale as the photochemical trapping by the neutral P_700_ [11]. Therefore, the quenching by P_700_^+^ can be an important photoprotective mechanism in PSI [12].

At cryogenic temperatures, the energy transfer dynamics in PSI are significantly altered. The thermal energy available for overcoming energy barriers is reduced, affecting the transfer efficiency from Red Chls, which act as excitation traps [13]. This is the main cause underlying the characteristic far-red emission that is widely used as a signature to probe PSI, both in vivo and in vitro [14,15].

The direct energy transfer between the Red Chls and RC becomes a critical pathway for de-trapping at low temperatures, which depends on the distance of the Red Chls to the RC, the orientation of their electronic transition dipoles, and their spectral properties [11,16]. An important factor is also the state of the RC (the reduced form P_700_, termed “open” RC, or the oxidized cation radical form P_700_^+^, termed “closed” RC), which strongly determines the spectral overlap and consequently the overall excitation kinetics. Akhtar et al. [17] studied the fluorescence kinetics of PSI in *Synechocystis* with open and closed RCs and showed that the far-red emission from the Red Chls decayed faster at 77 K when the RCs were open. This was explained by the larger spectral overlap between the Red Chls in *Synechocystis*, which emit at around 710 nm and P_700_, whereas the spectral overlap with the weakly absorbing P_700_^+^ was smaller. However, other cyanobacterial species boast significantly more red-shifted Chls that are spectrally farther from P_700_ but may, on the other hand, transfer energy to P_700_^+^, which has a weak broad absorption band at 820 nm typical for Chl cations. Shubin et al. [18] proposed that quenching by P_700_^+^ underlies the 77 K variable fluorescence of PSI. In support of this, Byrdin et al. [11] and Shibata et al. [19] have demonstrated that in *Thermosynechococcus vestitus* (*T. vestitus*, formerly *T. elongatus*) PSI, which has more red-shifted Chls with an emission maximum at 730 nm, the Red Chl emission is quenched more effectively when the RCs are closed—the opposite behavior of *Synechocystis*. Shibata further predicted energy transfer times from different putative Red Chls to P_700_. The efficiency of quenching by P_700_^+^ was found to vary widely among species, being the highest in species with the most red-shifted states, such as in *Arthrospira* sp. [16,18,20].

In this work, we investigate the excitation dynamics of trimeric PSI in the thermophilic cyanobacterium *Thermostichus vulcanus* (*T. vulcanus*) at 77 K using time-resolved fluorescence spectroscopy. The PSI in *T. vulcanus* has very similar spectroscopic properties to *T. vestitus*, as the two share the same amino acid sequence and three-dimensional structure [21]. We examine, in detail, the difference in the kinetics of the Red Chls when the RCs are in a closed state (P_700_^+^) or open state (neutral P_700_), showing that Red Chl emission is quenched in PSI with closed RCs. To gain further insight into the mechanism of quenching, we perform kinetic modeling of the 77 K experimental data and resolve different dynamics of excitation quenching depending on the energy of the Red Chls.

## 2. Results

### 2.1. Fluorescence Kinetics at 77 K

Time-resolved fluorescence measurements were performed at 77 K to resolve the dynamics of excitation and relaxation of the Red Chls. At this temperature, because of spectral line narrowing and the lack of available thermal energy, energy transfer from the Red Chls to the bulk antenna pigments is almost impossible, and excitations get trapped on the Red Chls. This leads to the well-known high fluorescence yield of PSI in the far-red wavelength range at 77 K (see Appendix A).

The fluorescence decay kinetics of PSI were measured in two different conditions: “closed-RC”, wherein the PSI RCs were pre-oxidized (P_700_^+^) by adding potassium ferricyanide under illumination before freezing, and “open-RC”, wherein the samples were frozen in complete darkness in the presence of ascorbate as an electron donor and phenazine methosulfate (PMS) as a mediator. In the latter case, the RCs are initially in the neutral state (“open”); however, the excitation light at low temperature creates long-lived radical pairs in a fraction of the PSI complexes [22]. If the recombination rate of these radical pairs is slower than the excitation rate, the RCs remain effectively closed during the fluorescence measurement. Therefore, it is not possible to measure PSI with fully open RCs but only with a mixture of about 50% [20,22].

Figure 1 compares the fluorescence decay kinetics and time-resolved fluorescence spectra of PSI with partially open RCs (P_700_ reduced) and with closed RCs (P_700_^+^) at selected emission wavelengths and decay times. More decay times are shown in Appendix A. The data show that the fluorescence at wavelengths shorter than 710 nm decays at approximately a similar rate in samples with oxidized and reduced RCs, whereas the fluorescence at longer wavelengths decays markedly faster in samples with oxidized RCs. These data suggest that the oxidized P_700_^+^ acts as a competent quencher for the far-red fluorescence, but the quenching efficiency depends on the emission wavelength.

For a quantitative description of the results, we performed a global multiexponential analysis of the fluorescence decays recorded in the range of 670–770 nm. Six exponential decay components provided a satisfactory fit of the fluorescence data under both oxidized/reduced conditions, as shown by the solid lines in Figure 1. The resulting lifetimes and decay-associated emission spectra (DAES) are plotted in Figure 2. Successive DAES have maxima that are gradually shifted to longer wavelengths as excitations are transferred from higher- to lower-energy pigments. In PSI with open RCs, the bulk antenna Chls emitting at wavelengths 680–690 nm decay with a lifetime of 19 ps, concomitantly populating the Red Chls, as evidenced by the negative amplitudes of the DAES at wavelengths longer than 700 nm. The next four DAES with increasing lifetimes from 75 ps to 1.2 ns have maxima around 715, 722, 730, and 740 nm, showing the existence of multiple Red Chl states that are both spectrally and kinetically distinct, as shown previously in *T. vestitus* [11,19]. The last component, with a 5.6 ns lifetime, has a very small amplitude (280-fold smaller than the first) and two peaks, indicating a mixture of free pigments and a very small fraction of extremely red-shifted Chls.

Under conditions of fully closed RCs, the global lifetime analysis revealed several notable changes in the fluorescence kinetics, specifically in the far-red region. The amplitude of the 1.2 ns component decreased nearly 20-fold, and the remaining amplitude of this component can be ascribed to a very low transient population in PSI with re-reduced RCs. The far-red peak in the 5-ns decay component also practically vanished. Conversely, the shorter-lived lifetime components (87 ps and 261 ps) gained amplitudes in the range of 730–740 nm. The analysis shows that the most red-shifted Chls have shorter fluorescence decay lifetimes when the RCs are closed, suggesting efficient excitation quenching by P_700_^+^. In contrast, Chls emitting at wavelengths below 720 nm are not affected by the state of the RC.

To further highlight the pronounced slowdown of fluorescence decay at longer wavelengths under closed RC conditions, we plotted the wavelength-dependent average lifetime for both open and closed states (Figure 3). The average lifetime remains nearly identical between 680 and 720 nm in both states. However, at emission wavelengths longer than 720 nm, the curves gradually diverge, with the average lifetime in the closed RC state becoming shorter than in open RCs. This suggests that the efficiency of excitation quenching is directly related to the spectral properties of the Red Chls, as expected if the mechanism of quenching involves direct Förster energy transfer to P_700_^+^.

### 2.2. A Target Analysis of the PSI Complex Under Open and Closed RC States

To understand the excitation dynamics of the Red Chls in more detail and extract biophysically relevant parameters, including rate constants of energy transfer and trapping, we performed kinetic modeling, or target analysis, simultaneously fitting the model to multiple fluorescence kinetic data recorded from PSI at 77 K. The basics of the simultaneous target analysis are described by van Stokkum et al. [23,24]. The kinetic model was constructed based on the initial six-component global lifetime analysis. Similarly, the model contained six distinct spectral components, of which one represented the bulk antenna Chls, four represented Red Chl states with progressively lower energies, as suggested by the DAES obtained from the global analysis (Figure 2), and an additional compartment was assigned mainly to free Chls.

Although it is mathematically possible to fit the data into a model where all four Red Chl species are connected in the same model, representing different Red Chl *a* sites, it is important to consider the effect of energetic disorder [25], i.e., multiple spectral components can be partly due to inhomogeneity of the Red Chl energies. Structure-based simulations of the 77 K fluorescence kinetics of PSI clearly demonstrated that the presence of inhomogeneously broadened Red Chls results in the appearance of multiple red-shifted DAES [17]. Following this, we used a heterogeneous kinetic model that allows for energy differences among the Red Chls (Figure 4). The heterogeneous kinetic scheme is similar to the model employed to describe the 77 K fluorescence kinetics of PSI in *Synechocystis* devoid of Photosystem II [26]. Individual PSI complexes are represented by three connected compartments. The bulk antenna Chls, along with the RC, are grouped together in a Bulk compartment (black in Figure 4) with a single effective decay rate representing trapping by the RC, either in a closed or open state. The Bulk compartment is connected to two Red Chl compartments (Red 1 and Red 2), representing different Red Chls states in the same PSI complex. The full model consists of three such tri-partite complexes, the other two (denoted as *b* and *c*) being kinetically and spectrally identical to the first one (*a*), except for the Red 2 compartment, which has varying species spectra and rate constants of energy transfer to and from the Bulk compartment. An additional unconnected compartment was included to account for free pigments, minor impurities, and inhomogeneities in the sample. It corresponds to the long-lived (5 ns) DAES in Figure 2.

The kinetic model was fitted simultaneously to the fluorescence kinetics of PSI with closed RCs and with a mixture of open and closed RCs. For simplicity, we assumed that the differences in the trapping rate constant between open and closed RCs were negligible (considering the limited time resolution), and we focused on the main experimentally observable differences, which were in the fluorescence decay rates of the Red Chls. Thus, the differences between open and closed RCs are modeled by varying the excitation decay rate of the Red 2b and Red 2c compartments (Figure 4A), assuming that in PSI with closed RCs, the decay is faster due to excitation quenching by P_700_^+^. The spectrum of the long-lived unconnected compartment was also allowed to vary between datasets since the addition of the electron donor PMS might have induced the detachment of the Chls in a small fraction of the complexes.

Figure 4 summarizes the fitting results, showing the rate constants of energy transfer and trapping, the species-associated emission spectra (SAES), and the calculated time-dependent concentrations. The initial concentrations and the matrices of amplitudes in the different types of modeled PSI complexes are shown in Appendix A (Appendix A). The model fits the experimental fluorescence kinetics satisfactorily, including the kinetics of the Red Chls in PSI with either closed or open RCs, as shown in Figure 5 for two characteristic wavelengths (more traces are shown in Appendix A) [22,27].

The estimated rate constant of trapping is 46 ns^−1^ (the decay rate of the Bulk(+RC) compartment). The SAES of the Bulk compartment (black in Figure 4) shows that it also contains some red-shifted forms, which are populated faster (4–6 ps) than the time resolution of the experiment [28,29,30]. The area of this compartment is smaller than that of the excitonic Red Chl *a* species; however, since the decay lifetime is shorter than the IRF, the amplitudes may be inaccurate. The negative amplitudes in the reconstructed DAES, as well as the negative amplitudes in the amplitude matrices (Appendix A), indicate the downhill EET. The main downhill EET timescale is 13–15 ps in all complexes. The secondary downhill EET to the most red-shifted Chl *a* occurs with a lifetime of ≈109 ps.

The Gibbs free energies (relative to Bulk) of the Red Chl compartments, Red 1, 2a, 2b, and 2c, are 8.6, −9.3, −16, and −21 meV (calculated from the ratios of the forward and backward energy transfer rate constants). The progression in the free energy difference is in agreement with the gradual shift of the corresponding SAES to longer wavelengths. It is important to note that the well-resolved SAES fit both experimental conditions (open/closed RC) except for the additional unconnected component. A zero-constraint above 720 nm was applied for the Bulk compartment’s SAES. The constraint causes a small increase in the fit error (RMS) but negligible differences in the fit traces (Figure 5 and Appendix A). The SAES of a very small amount of long-lived impurity (grey in Figure 4B,C) peaked at 680 nm (attributed to free Chl *a*) and showed a minor peak around 740 nm (attributed to a small amount of Red Chl that cannot transfer its energy because it is too far away from the closed RC).

The three types of complexes (*a*, *b*, and *c*) have relative fractions of 33 ± 5, 41 ± 5, and 26 ± 5%, respectively. However, the fraction of open RCs in PSI with added electron donors is most difficult to estimate. In bulk preparation, the measurable fraction of open RCs is expected to be approximately 50% [20,22]. We further consider connectivity between protomers in the trimeric PSI complexes, that is, if Red Chl excitations in one of the protomers can be quenched not only by P_700_^+^ in the same protomer but also by P_700_^+^ in an adjacent one [27]. In line with this, a non-linear relationship was reported between the fraction of closed RCs and the quenching of 760 nm fluorescence of trimeric PSI in *Arthrospira* sp. In the extreme case, where quenching by any of the protomers is equivalent, the ‘effective’ fraction of closed RCs is equal to the fraction of PSI trimers containing at least one oxidized RC. In the following, we show modeling results for an intermediate ‘effective’ fraction of 70% closed RCs.

The most notable kinetic difference in PSI complexes with closed RCs is the faster decay of the red-shifted forms, with the most red-shifted (Red 2c) being the most strongly quenched, with a rate constant of 6 ns^−1^. Next, Red 2b is mildly quenched, with a decay rate constant of 0.74 ns^−1^. The other compartments have natural decay rates of 0.19 ns^−1^. These decay pathways result in the faster depopulation of the Red 2b and 2c forms, visible in the time-dependent concentration plots (orange and magenta in Figure 4C). From the amplitude matrices (Appendix A), we see that the PSI complex ‘*c*’ with open RCs has a long decay lifetime of 1.38 ns. This decreases to 155 ps in the same complex with a closed RC. Analogously, the open complex ‘*b*’ has a lifetime of 615 ps, and this decreases to 463 ps in the closed complex. Thus, a modest concentration of fully open PSI RCs (30%) explains the observed increase in the observed lifetimes. The Red 2*a* compartment (in PSI complex ‘*a*’) decays with a lifetime of 310 ps due to uphill EET and trapping by the RC, independently of the RC state [11,31]. Note that a similar result was obtained when the effective fraction of closed RCs in the sample frozen with electron donors was set to 50%. The most notable difference in this case was that the highest quenching rate constant (for the Red 2c compartment) was 7 ns^−1^.

## 3. Discussion

The fluorescence kinetics of PSI in *T. vulcanus* recorded at 77 K show unequivocally that the redox state of the RC has a profound effect on the excitation lifetime of the lowest energy Red Chls. The wavelength dependence of the average fluorescence lifetimes of PSI with closed RCs suggests that the more red-shifted forms have faster excitation decay. This is consistent with a mechanism wherein excitations residing in the red-shifted Chls are quenched via Förster resonant energy transfer to the oxidized RC P_700_^+^, followed by rapid nonradiative deactivation of the excited P_700_^+^* [32,33]. Energy transfer is enabled by the spectral overlap between the exciton emission and the absorption of the Chl cation radical. Hence, the quenching is most effective for states emitting at wavelengths closer to the absorption maximum of P_700_^+^, around 820 nm [34].

The simultaneous target analysis explains the different fluorescence kinetics of PSI with open and closed RCs at 77 K by invoking the above mechanism. Introducing a quenching rate constant to the red-shifted Chls in PSI with closed RC and no other changes fully reproduced the fluorescence data. The analysis confirmed quantitatively that the emission maxima of the Red Chls and the quenching rate are positively related and revealed that only states with sizeable emission at wavelengths longer than ≈730 nm are efficiently quenched. Conversely, the decay lifetime of intermediate states with emission maxima around 710–720 nm was independent of the state of the RC. Note that these states still decay with lifetimes much shorter than the free Chl decay lifetime of ≈5 ns; however, in this case, the primary de-excitation pathway is via uphill energy transfer to higher-energy Chls followed by trapping by the RC.

The target analysis convincingly resolves at least four spectral forms of red-shifted Chls that are kinetically and spectrally distinct, in accordance with several previous studies [17,19,26]. Each of the Red Chl compartments likely represents a state contributed by multiple excitonically interacting Chl *a* pigments. The broad emission spectra are consistent with the states having a charge-transfer character [35,36]. Most likely, the apparently wide range of energies (and consequently kinetics) of the Red Chls is a result of a high degree of structural inhomogeneity and resulting energetic disorder. This is consistent with both theoretical quantum chemical/molecular dynamics simulations [25] and single molecule spectroscopy measurements [37], showing the dynamic and heterogeneous nature of red-shifted Chls in PSI. 

The different pathways of de-trapping of the Red Chl excitations are illustrated in Figure 6 using the structure of PSI in *T. vestitus.* Three groups of Chls are denoted as Red Chls—A_32_/B_7_, B_37_/B_38_, and B_31_/B_32_/B_33_ [17]. In this tentative assignment, the A_32_/B_7_ state has an intermediate energy level and is de-trapped primarily via uphill energy transfer to bulk Chls, denoted with a curved dashed arrow, similar to the Red 1 compartment in the model, which decays with a 108 ps lifetime. The B_37_/B_38_ energy has a longer emission wavelength that poorly overlaps with the bulk Chls and with the RCs, resulting in slow de-trapping. The most red-shifted state, presumably contributed by the B_31_/B_32_/B_33_ group, is most effectively quenched by direct transfer to P_700_^+^. Evidently, the lifetimes and the pathways (indirect or direct de-trapping) can vary following the variation in Chl energies in individual PSI complexes.

Our findings offer insight into the excitation dynamics and energetic heterogeneity of the Red Chls in PSI in *T. vulcanus* and their direct and indirect interaction with the RC. The quantitative parameters obtained by the simultaneous target analysis provide a framework for a detailed understanding of the experimental time-resolved spectroscopy data at 77 K. The quenching rates could be used as a benchmark for structure-based models of the excitation dynamics—if the rate of quenching is determined by the Förster energy transfer to the RC, it is, in turn, related to the distance to the RC [19]. However, the main implication of the heterogeneous model employed here is that the relatively wide range of energies adopted by the Chls is the dominant factor determining the kinetics rather than their position within the complex, as was suggested by Russo et al. [9] for higher-plant PSI. The same could be true for other systems possessing low-energy traps. An active area of current research is the light harvesting in PSI containing Chl f, which has been proposed to be located at the periphery of the antenna, closer to the RC Chls, or within the RC itself [38,39,40]. In this case, the thermodynamic factor would be even more impactful on the kinetics and efficiency of excitation energy utilization, and the slowdown of overall trapping can largely be explained with slow uphill energy transfer from the low-energy Chl f but the structural factors are yet to be elucidated [40,41].

The results presented here highlight the marked effect of the far-red fluorescence yield of PSI on the redox state of the RC and clarify, in a quantitative manner, the direct interactions between the Red Chls and the RC, shedding new light on the origins of the widely used low-temperature fluorescence as a tool to probe both photosystems. 

## 4. Materials and Methods

### 4.1. Sample Preparation

The thermophilic cyanobacterium *T. vulcanus* was grown photoautotrophically as a batch culture in BG11 medium (pH 7.5) at 45 °C under continuous illumination with a white, fluorescent lamp at an intensity of 50–100 μmol photons m^−2^ s^−1^ photon flux density. The PSI samples were isolated from freshly prepared thylakoid membranes following the protocol of Fromme and Witt [42] with small modifications. The thylakoid membranes isolated from one-week-old cells were solubilized by incubating with 2% n-dodecyl β-D-maltoside (β-DDM) at 4 °C for 30 min in the dark. The suspension was centrifuged for 15 min at 10,000× *g* to remove the insolubilized material. The supernatant was then loaded on a stepwise (6 steps, 0.2–0.9 M) sucrose gradient containing 20 mM HEPES (pH 7) and 0.05% of β-DM, followed by centrifugation at 220,000× *g* for 17–18 h at 4 °C. The gradient fractions containing the PSI trimers were collected by a syringe, washed in a medium containing 0.03% β-DDM, concentrated using Amicon Ultra filters (Millipore, Burlington, MA, USA), characterized by absorption and steady-state fluorescence spectroscopy (Appendix A), and stored at −80 °C until use.

The samples were suspended in 20 mM Tricine buffer (pH 7.5) and 0.03% β-DDM for spectroscopic characterization. For the measurements at 77 K, the buffer was supplemented with 60% glycerol.

### 4.2. Time-Resolved Fluorescence Spectroscopy

Picosecond time-resolved fluorescence measurements were performed with a time-correlated single-photon counting (TCSPC) instrument (FluoTime 200/PicoHarp 300 spectrometer, PicoQuant, Berlin, Germany). The source of excitation was a Fianium WhiteLase Micro (NKT Photonics, Southampton, UK) supercontinuum laser, providing white-light pulses at a repetition rate of 20 MHz centered at 440 nm. The total instrument response (IRF) width was 40 ps, measured using 1% Ludox as a scattering solution. All the samples were diluted to an absorbance of 0.03 at the excitation wavelength. The suspension was placed in a 1 mm demountable cryogenic quartz cell and cooled in an optical cryostat (Optistat DN, Oxford Instruments, Abingdon, UK). The fluorescence decays were recorded at wavelengths of 670–770 nm with 5 or 10 nm steps. Measurements were performed under two conditions: partially open RCs and closed RCs. For measurements with closed (oxidized) RCs, 1 mM potassium ferricyanide was added, and the sample was pre-illuminated with white light for 5 min before freezing in liquid nitrogen. For partially open RCs, dark-adapted samples were frozen in the presence of 40 μm PMS and 20 mM sodium ascorbate, and the excitation light intensity was reduced by 20-fold compared to the closed state. The estimated spectra have been corrected for the spectral response of the detector. Global multiexponential lifetime analysis with IRF reconvolution was performed using MATLAB (version 2024a). Target analysis was performed according to van Stokkum et al. [23,24] using an equal-area constraint of the SAES to estimate the equilibria [43].

## Figures and Tables

**Figure 1 ijms-25-12430-f001:**
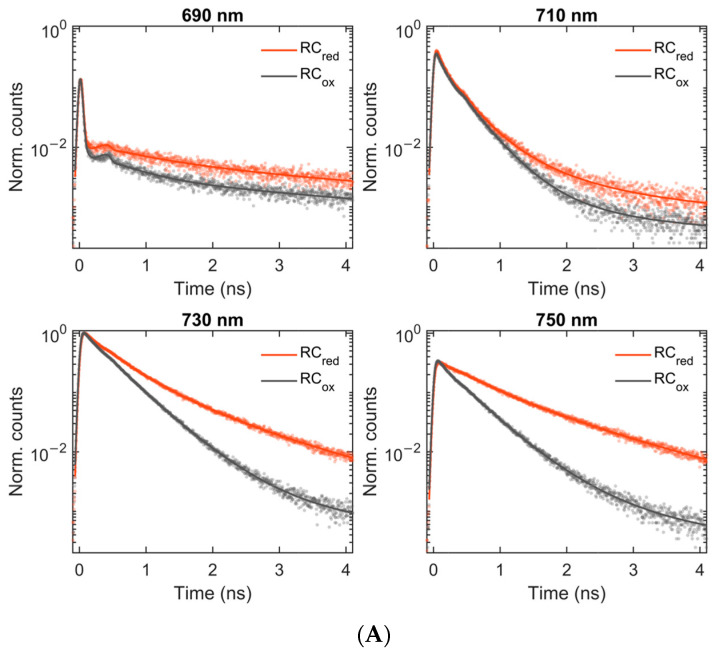
The time-resolved fluorescence of PSI in *T. vulcanus* at 77 K. (**A**) The fluorescence decay kinetics at four emission wavelengths from 690 nm to 750 nm. The symbols indicate the measured data points, and the lines are obtained by global multiexponential fitting. RC_red_ represents a sample containing ≈50% open (reduced) RCs, and RC_ox_ represents a sample with closed (oxidized) RCs. The decays are normalized at the respective global maximum and plotted on a semi-logarithmic scale. (**B**) The time-gated fluorescence spectra at four decay times from 0.1 to 2 ns. The symbols and lines represent data and fit, respectively.

**Figure 2 ijms-25-12430-f002:**
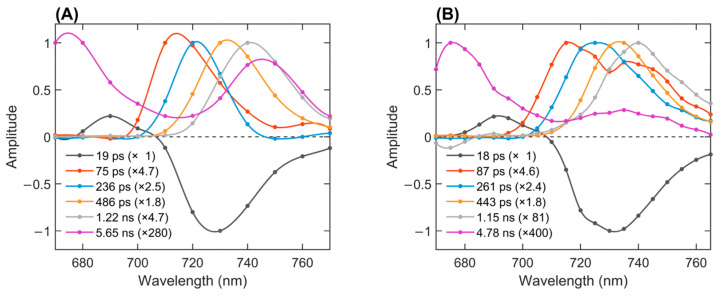
The decay-associated fluorescence emission spectra of isolated PSI complexes obtained from a global analysis of fluorescence decays recorded at 77 K with 440 nm excitation. (**A**) PSI with open and closed RCs (≈50% P_700_^+^) and (**B**) PSI with closed RCs (P_700_^+^). The spectra are normalized to their maxima with the relative normalization factors shown in parentheses.

**Figure 3 ijms-25-12430-f003:**
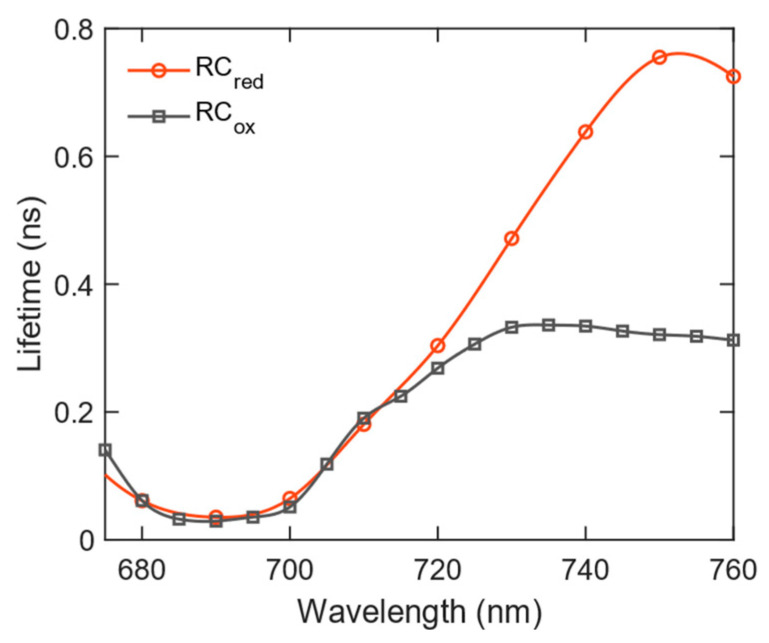
The wavelength dependence of the amplitude-weighted average fluorescence lifetime $\tau_{av}=\sum_{i}a_{i}\tau_{i}/\sum_{i}a_{i}$ at 77 K, compared for the open and closed RCs. Note that decay components with an ≈5-ns lifetime attributed to free Chls are excluded from the calculation.

**Figure 4 ijms-25-12430-f004:**
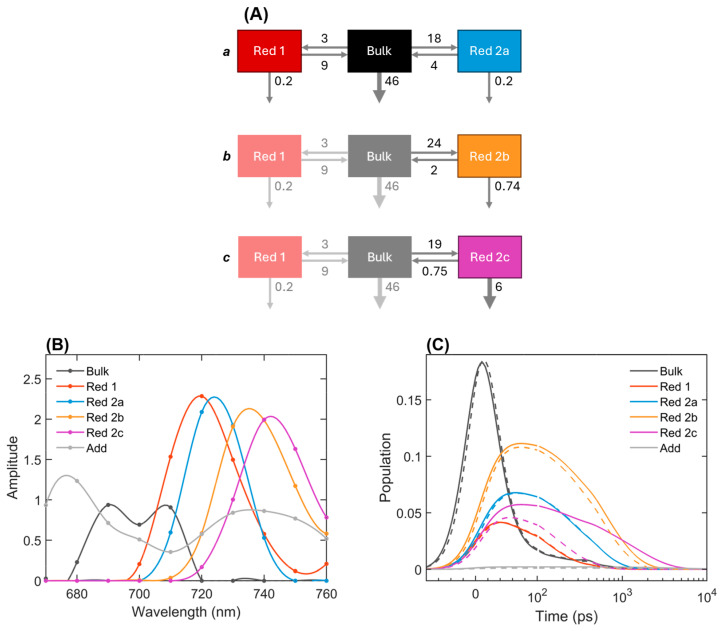
The heterogeneous target analysis of PSI kinetics at 77 K. (**A**) The kinetic model schemes of PSI with a closed RC (P_700_^+^) with energy transfer and trapping rate constants in ns^−1^. With an open RC, the decay rates of Red 2b and Red 2c are both 0.2 ns^−1^. (**B**) The species-associated emission spectra (SAES), including an unconnected compartment “Add”. The amplitude of the “Bulk” SAES is multiplied by 5 for readability. (**C**) The species transient populations (time-dependent concentrations). The solid/dashed lines represent open/closed RCs. Note that the horizontal scale is linear until 100 ps and logarithmic thereafter.

**Figure 5 ijms-25-12430-f005:**
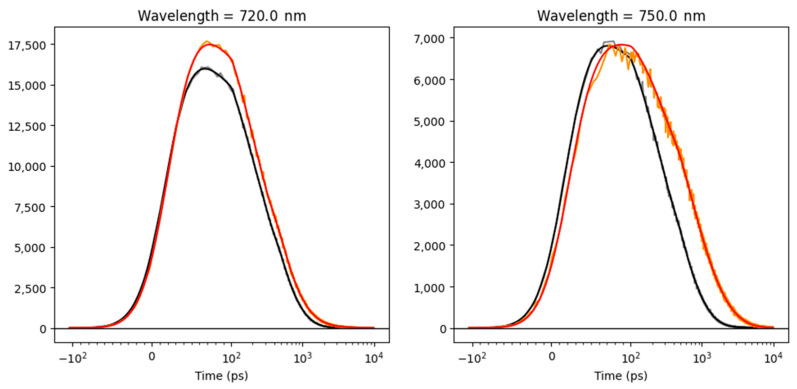
The time traces of the PSI emission at two wavelengths (indicated in the title of the panels) after 440 nm excitation at 77 K. The grey (orange) and black (red) lines indicate the data and the target analysis fit of the PSI in the closed and “open” RC experiments, respectively. Note that the time axis is linear until 100 ps and logarithmic thereafter. Note also that each panel is scaled to its maximum. The overall rms error of the fit was 18.3.

**Figure 6 ijms-25-12430-f006:**
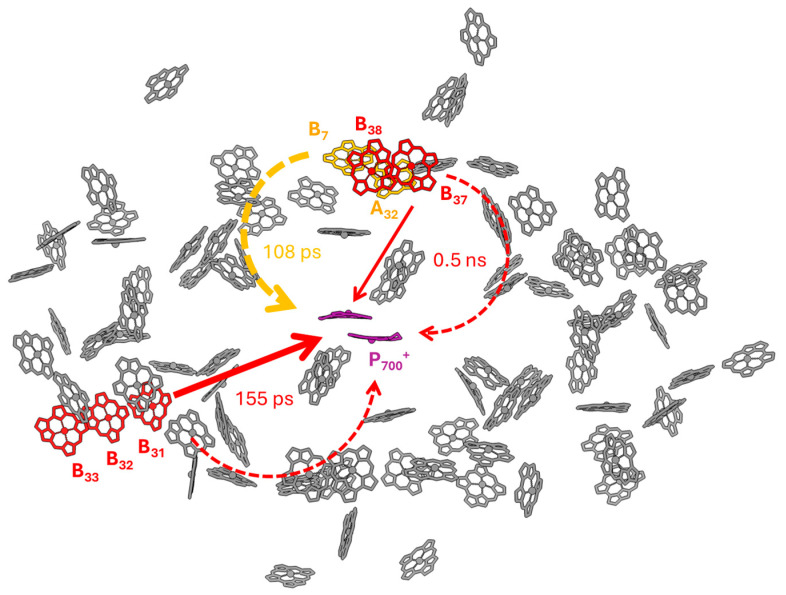
An illustration of the quenching of Red Chl emission in *T. vulcanus’s* PSI based on the structure and tentative assignment of Chls contributing to the low-energy states. The Chl positions are drawn using PDB 1JB0 [2]. Red Chls are denoted with yellow and red colors. The curved dashed arrows represent multistep energy transfer to the RC via intermediate-energy bulk Chl states. The straight solid arrows represent direct Förster resonance energy transfer to the oxidized RC (P_700_^+^). The transfer lifetimes can vary depending on the (inhomogeneously distributed) Red Chl energies. The intermediate-energy state contributed by Chls A_32_/B_7_ is de-trapped mainly by uphill energy transfer through the bulk Chls. The low-energy state contributed by Chls B_31_/B_32_/B_33_ is quenched by direct transfer to P_700_^+^.

## Data Availability

The raw data supporting the conclusions of this article will be made available by the authors upon request.

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
