# Peer review of "The Quenching of Long-Wavelength Fluorescence by the Closed Reaction Center in Photosystem I in Thermostichus vulcanus at 77 K"

_ijms, 2024, doi:10.3390/ijms252212430_

Round 1

Reviewer 1 Report

Comments and Suggestions for Authors

The paper presents a study of the photosystem I (PSI) trimer from T. vulcanus using time-resolved fluorescence spectroscopy. The goal is to compare open and closed reaction centers (reduced/oxidized P700) and identify the spectroscopic and kinetic behavior of red chlorophylls in PSI. The kinetic analysis is presented in detail, a plausible kinetic model is described, and its functional implications are clearly discussed. A profound effect of the redox state of the RC on the excitation lifetime is documented, with interpretation of general and specific observations being offered in terms of two possible pathways (red Chls -> P700+ versus Red -> higher-E Chl -> RC) as well as structural inhomogeneity. Overall this is solid research work that is presented clearly and competently.

I have only two minor remarks that can be considered: 1) page 3: it is commented that it is not possible to measure PSI with fully open RCs: can the percentage of open/closed be more accurately known, or somehow quantified in each measurement (e.g. by EPR?) and does the assumption regarding this value affect the kinetic modeling? 2) Can it be stated clearly whether the four Red Chl states correspond to (four) distinct pigments or if they can be domains of more than one Chl each? 3) Is there any evidence, or can there be an interpretation of the data to accommodate the existence of low-energy antenna-localized charge-transfer states?

Author Response

Comment 1: page 3: it is commented that it is not possible to measure PSI with fully open RCs: can the percentage of open/closed be more accurately known, or somehow quantified in each measurement (e.g. by EPR?) and does the assumption regarding this value affect the kinetic modeling?

Response 1: We do not have the capability to measure the exact number of open/closed RC during the fluorescence measurements but have based our analysis on literature data under similar conditions (specifically refs 18-20). Note however, that the data clearly show the efficient quenching of the far-red emission in oxidized-RC conditions compared to reduced-RC conditions. This is now illustrated more prominently in the new Fig. 1 and Fig. S2.

The reviewer is correct that the assumed ratio of open/closed RCs might affect the kinetic modeling. We have done the model fitting assuming different values and have obtained a broadly similar result – the quenching rate constant of the redmost Chl changes to 6 or 7 /ns assuming 50% or 70% closed RCs respectively. We have extended the results section to reflect this.

Comment 2: Can it be stated clearly whether the four Red Chl states correspond to (four) distinct pigments or if they can be domains of more than one Chl each? 3) Is there any evidence, or can there be an interpretation of the data to accommodate the existence of low-energy antenna-localized charge-transfer states?

Response 2: The red Chl a compartments most likely represent low-energy excited states to which multiple excitonically interacting Chl a pigments participate. Previous experimental data from several groups have indeed suggested that these states have a mixed excitonic and charge-transfer character. This is now mentioned in Discussion (line 300).

Reviewer 2 Report

Comments and Suggestions for Authors

Comments paper “Quenching of the long-wavelength fluorescence by the closed reaction centre of Photosystem I of Thermostichus vulcanus at 77 K”

Akhtar et al., 2024, IJMS, 3292251

Long wavelength or “Red” chlorophylls (Chls) occur in most organisms. In this paper,  a study investigating the excitation dynamics of Red Chls in Photosystem I of a cyanobacterium at cryogenic temperatures is performed and examines the role of oxidized RC in modulating the fluorescence kinetics. The kinetics of red Chls for samples with open and closed RCs are recorded using time-resolved fluorescence spectroscopy. The results highlight the marked effect of the far-red fluorescence yield of PSI on the redox state of the RC and gives insight in the direct interactions between Red Chls and the RC.

The paper is well written, the different sections are quite good understandable and the figures are taken care off. However, there are some minors remarks and suggestions to improve the manuscript.

p.3, line 110-111: Why should this be 50% and not 30 or 70%? What is the rationale behind the 50%? Is this based on the study of Schlodder et al, see ref. 19?

p.5, line 192: Cannot find this "Add" in fig 4 A, but it is present in B and C.

p.6, legend figure 4: The number of 0.2 ns-1 is indicated for Red 2a, but is for Red 2b 0.74 and 6 for Red 2c! Is this correct?

Discussion:

In order to understand the energy transfers and the impact of the heterogeneity  of the red-shifted Chls, it would be handsome and more informative using schemes and/or drawings to make the text better understandable.

Author Response

Thank you for the dedicated work and helpful comments! Below are our point-by-point responses:

Comment 1: p.3, line 110-111: Why should this be 50% and not 30 or 70%? What is the rationale behind the 50%? Is this based on the study of Schlodder et al, see ref. 19?

Response 1: Indeed, the ~50% closed RC in the sample with electron donors is taken from Schlodder et al. 1998 and 2005 (refs. 18, 20). The citations are added to the sentence. The rationale for estimating the ratio closed/open RCs is now further discussed in the Results section (page 7, lines 250-260).

Comment 2: p.5, line 192: Cannot find this "Add" in fig 4 A, but it is present in B and C.

Response 2: The reference to “Add” was moved to the figure legend.

Comment 3: p.6, legend figure 4: The number of 0.2 ns-1 is indicated for Red 2a, but is for Red 2b 0.74 and 6 for Red 2c! Is this correct? 

Response 3: The dissipation rate constants represent not only the natural excited-state decay (0.2 ns-1) but also the direct quenching by the oxidized RC. The more red-shifted Chls are more effectively quenched by the RC, hence the faster decay up to 6 ns-1. This is now illustrated in a more intuitive way in Figure 6.

Comment 4: In order to understand the energy transfers and the impact of the heterogeneity of the red-shifted Chls, it would be handsome and more informative using schemes and/or drawings to make the text better understandable.

Response 4: Thank you for the suggestion. We have included a new figure 6, that illustrates the different pathways of de-trapping (indirect and direct) of different red Chls based on the PSI structure and the kinetic modelling results.  Additional text describing the figure is included in Discussion, lines 310-320.

Reviewer 3 Report

Comments and Suggestions for Authors

Review

Quenching of the long-wavelength fluorescence by the closed reaction centre of Photosystem I of Thermostichus vulcanus at 77 K

Parveen Akhtar, Ivo H.M. van Stokkum, and Petar H. Lambrev

The manuscript by P. Akhtar et al. presents time-resolved 77K chlorophyll fluorescence measurements on Photosystem I extracted from Thermostichus vulcanus. It shows that the fluorescence of strongly red shifted chlorophylls is quenched by the presence of the oxidized reaction center P700+. The data is fitted and explained with a reaction model of inhomogeneous samples with distinct red shifted chlorophyll centers. From this, the authors conclude that some of the excited state red chlorophylls are quenched by direct Förster energy transfer to P700+ and suggest that the red chlorophyll’s arrangement allows for a high degree of structural inhomogeneity.

In general, the manuscript is well written and especially the data analysis is well performed. Although similar measurements have been performed by other groups on other organisms before, the study adds to the field by analyzing a different organism and performing the kinetic reaction target analysis. The former is especially important, as the red shifted chlorophylls differ strongly depending on the organism, as also pointed out in the introduction.

I therefore recommend the manuscript for publication.

Please find below some minor points, I want the authors to consider.

Main

Results

·      Fig1, fig2 raw decay spectra and fit quality
To allow the reader to judge about the raw data and fit quality, an extra figure should be added to the SI showing the spectral decay at selected times and the fit on those spectra. The fit could also be shown in figure 1.
I want to note that I have had no access to the supplementary information: in the main text fig S2 is said to show the fit for the target analysis – the same should be presented for the exponential fit if it is not already.

Minor

Introduction

·      P2, l53, abbreviation
PSII first time mentioned: Photosystem II (PSII)

·      P2, l67, needs clarification or citation
It is stated that ‘the direct energy transfer from the red Chls to the RC becomes a critical pathway’. This may need a citation excluding the uphill and therefore less likely but not impossible energy transfer

·      P2, l81, renaming
also p4, l140

o   To the best of my knowledge, T. elongatus has been renamed to T. vestitus; maybe the authors want to add the new name as well

·      P2, last paragraphs, T. vulcanus

o   It may be nice to note the similarity / differences of red Chls of T. vulcanus compared to other species, especially T. vestitus

Materials and Methods

·      P9, l343
FWHM of fluorescence measurements

o   Which filter or monochromator setting was used? What was the spectral FWHM of probed fluorescence in nm?

Results

·      P3, l114, 710 nm decay equal with P and P+

o   In the introduction the authors refer to the study on Synechocystis (which one?) and note that it was found that the 710 nm fluorescence decayed faster with open RC. How is the here found independence of the 710 nm comparing to the Synechocystis study?

·      P5, l168-194, 
inhomogeneous vs homogeneous model

o   Although up to the authors, the results of a homogenous model with three different red chl clusters would be interesting to see as there are several candidates for red chlorophyll clusters within one PSI monomer

o   Or it could be stated more clearly why the homogeneous model was completeley ruled out

·      Fig.4 C
it may be helpful to note what the dashed and the full line represent

·      P7, l263
the sentence refers to time-dependent concentration plots, the reference is to figure 4B. The authors could add 4C

Discussion

·      P7, l 277
What is the 2nd part of the mechanism – rapid nonradiative decay of P700+* – based on – may need a citation? The here presented study only probes the fluorescence up to 760 nm while the fluorescence of P700+* is likely red shifted of the maximum around 820 nm, so it likely wasn’t probed.

Author Response

Thank you for the dedicated work and helpful comments! Below are point-by-point responses.

Comment 1: Fig1, fig2 raw decay spectra and fit quality
To allow the reader to judge about the raw data and fit quality, an extra figure should be added to the SI showing the spectral decay at selected times and the fit on those spectra. The fit could also be shown in figure 1.

Response 1: Thank you for the helpful suggestion. We have changed Fig. 1 to show both the measured decays as symbols and exponential fits as solid lines for six different wavelengths. Additionally, time-gated spectra are shown in Fig. 1B and supplementary Fig. S2.

Comment 2: in the main text fig S2 is said to show the fit for the target analysis – the same should be presented for the exponential fit if it is not already.

Response 2: The fit is now shown in Fig. 1 and new Supplementary Fig. S2.

Comment 3: P2, l53, abbreviation
PSII first time mentioned: Photosystem II (PSII)

Response 3: Corrected.

Comment 4: P2, l67, needs clarification or citation
It is stated that ‘the direct energy transfer from the red Chls to the RC becomes a critical pathway’. This may need a citation excluding the uphill and therefore less likely but not impossible energy transfer

Response 4: We have added the missing citations [11, 16].

Comment 5: To the best of my knowledge, T. elongatus has been renamed to T. vestitus; maybe the authors want to add the new name as well

Response 5: We have used the new name and referred to the old name in the first mention.

Comment 6: P2, last paragraphs, T. vulcanus
It may be nice to note the similarity / differences of red Chls of T. vulcanus compared to other species, especially T. vestitus

Response 6: T. vulcanus PSI is practically identical to T. vestitus owing to the 100% identical aminoacid sequence of their major subunits. We have now mentioned this in the introduction (line 92): “PSI of T. vulcanus has very similar spectroscopic properties to T. vestitus, as the two share the same aminoacid sequence and three-dimensional structure [21].”

Comment 7: P9, l343
FWHM of fluorescence measurements
Which filter or monochromator setting was used? What was the spectral FWHM of probed fluorescence in nm?

Response 7: The fluorescence emission wavelength was selected using a Czerny-Turner monochromator with a linear dispersion of 5 nm/mm at 600 nm. We used 1 mm entrance and exit slits; therefore, the spectral resolution (FWHM) was about 5 nm. The sentence was added (Materials and Methods, line 374): “The spectral bandwidth of the monochromator was 5–6 nm.”

Comment 8: P3, l114, 710 nm decay equal with P and P+

In the introduction the authors refer to the study on Synechocystis (which one?) and note that it was found that the 710 nm fluorescence decayed faster with open RC. How is the here found independence of the 710 nm comparing to the Synechocystis study?

Response 8: We agree that this is a valid question. One would expect that similarly positioned red Chls with similar spectra should exhibit the same behaviour in both species. In the Synechocystis study [17], states with maxima at 710 nm decayed slightly faster in samples with open RCs, whereas the emission at 710 nm in the present study had a similar decay rate in open/closed RCs. However, it should be considered that the spectra of the Red Chls are remarkably broad and largely overlapping. In this case, a small ‘negative’ quenching at 710 nm will be masked by the much stronger quenching at longer wavelengths.

Comment 9: P5, l168-194, 
inhomogeneous vs homogeneous model

o   Although up to the authors, the results of a homogenous model with three different red chl clusters would be interesting to see as there are several candidates for red chlorophyll clusters within one PSI monomer

o   Or it could be stated more clearly why the homogeneous model was completeley ruled out

Response 9: We have tested homogeneous models to fit the kinetics, however, they were ruled out for two reasons: 1) They did not provide physically plausible spectra or thermodynamic parameters (free energy difference); 2) it is established that the red chls have substantial inhomogeneous broadening which will be manifested as the appearance of multiple spectral components. We have extended the text to reflect that although it is mathematically possible to use a homogeneous model, a heterogeneous one is more realistic (lines 185-195).

Comment 10: Fig.4 C
it may be helpful to note what the dashed and the full line represent

Response 10: Amended.

Comment 11: P7, l263
the sentence refers to time-dependent concentration plots, the reference is to figure 4B. The authors could add 4C

Response 11: Amended.

Comment 12: P7, l 277
What is the 2nd part of the mechanism – rapid nonradiative decay of P700+* – based on – may need a citation? The here presented study only probes the fluorescence up to 760 nm while the fluorescence of P700+* is likely red shifted of the maximum around 820 nm, so it likely wasn’t probed.

Response 12: We have added relevant citations [31, 32].

Reviewer 4 Report

Comments and Suggestions for Authors

In this study, the authors investigate the excitation dynamics of red chlorophylls in photosystem I of the cyanobacterium Thermostichus vulcanus at cryogenic temperatures (77 K) and compare their fluorescence kinetics associated with open and closed reaction centers (RCs). They describe a model for the fluorescence quenching of red Chls by the oxidized RCs, P700+ as a PSI photoprotective mechanism. The study is interesting and provides important insight into PSI exciton energy transfer. I have only one comment that the authors should address before publication: The main conclusion of the study is that the relatively wide range of energies adopted by the Chls is the dominant factor determining the kinetics, rather than their position within the complex. Could this be a general concept? Might it depend on the photosynthetic organism? For example, in Nat Commun 11, 238 (2020). https://doi.org/10.1038/s41467-019-13898-5 or BBA - Bienergetics 2020, 1861 (8), 148206 the position of the far-red Chl f  functions to enhance the uphill energy transfer to the RCs. Despite the fact that Chl f is a different form compared to the red shifted Chls a in PSI, I would expect a general adaptation for the red shifted traps forming around PSI. Why might different photosynthetic organisms use different strategies for uphill energy transfer to the RCs or photoprotection?

Author Response

Thank you for evaluating the manuscript and for the helpful comment that gives a broader perspective on the results!

Comment 1: The main conclusion of the study is that the relatively wide range of energies adopted by the Chls is the dominant factor determining the kinetics, rather than their position within the complex. Could this be a general concept? Might it depend on the photosynthetic organism? For example, in Nat Commun 11, 238 (2020). https://doi.org/10.1038/s41467-019-13898-5 or BBA - Bienergetics 2020, 1861 (8), 148206 the position of the far-red Chl f  functions to enhance the uphill energy transfer to the RCs. Despite the fact that Chl f is a different form compared to the red shifted Chls a in PSI, I would expect a general adaptation for the red shifted traps forming around PSI. Why might different photosynthetic organisms use different strategies for uphill energy transfer to the RCs or photoprotection?

Response 1: The parallel with Chl f is relevant insofar Chl f also serves as a low-lying state that has an impact on the excitation dynamics of PSI. Although this is currently an active area of research, it can be inferred from recent modeling studies on PSI with Chl f that the thermodynamic factor (slow uphill energy transfer) has a leading role in the overall kinetics but the structural factors are yet to be elucidated (Tros et al., Chem, 2021; van Stokkum et al., iScience, 2023). We have now included this parallel in the manuscript, page 9 lines 360-370.